# 'It is a disease which comes and kills directly': What refugees know about COVID-19 and key influences of compliance with preventive measures

Adelaide M. Lusambili [1]*, Michela Martini [2], Faiza Abdirahaman[2], Asante Abena[2], Joseph N. Guni[1], Sharon Ochieng[1], Stanley Luchters[1,3,4]

1 Department of Population Health, Medical College, Aga Khan University, Nairobi, Kenya, 2 International Organisation of Migration (IOM), United Nations, Nairobi, Kenya, 3 Department of Public Health and Primary Care, International Centre for Reproductive Health, Ghent University, Ghent, Belgium, 4 Department of Epidemiology and Preventive Medicine, Monash University, Melbourne, Australia

* adelaide.lusambili@aku.edu

**Data Availability Statement:** Due to ethical reasons, data from this study are unavailable. We interviewed antenatal and postnatal women

## Abstract

### Background

Refugees are at increased risk for COVID-19 infection in part due to their living conditions, which make it harder to adopt and adhere to widely accepted preventive measures. Little empirical evidence exists about what refugees know about COVID-19 and what they do to prevent infection. This study explored what refugee women and their health care workers understand about COVID-19 prevention, the extent of their compliance to public health recommendations, and what influences the adoption of these measures.

### Methods

In October 2020, we conducted 25 in-depth interviews with facility and community health care staff (n = 10) and refugee women attending antenatal and postnatal care services (n = 15) in Eastleigh, Nairobi.

### Findings

While researchers found a high level of awareness about COVID-19 and related prevention and control measures among refugee women, various barriers affected compliance with such measures, due in part to poverty and in part to rampant misconceptions informed by religious beliefs and political narratives about the virus.

### Conclusions

These findings indicated that Kenya's Ministry of Health needs to institute a concerted and continuous education program to bring refugee communities up to speed about COVID-19 and its prevention. In addition to disseminating information about the need to wear masks and repeatedly wash hands, supplies—masks, soap and access to water—need to be

alongside their health care workers. The sample size was small, and the participants came from one study site. Because of their roles, readers can easily deduce from the transcripts who the participants are. Due to the sensitivity of refugees' experience in terms of their legal status and treatment, we cannot share data. To access this data, contact the Aga Khan University Institutional Ethics committee at AKUKenya. researchoffice@aku.edu or the study principal investigator at adelaide.lusambili@aku.edu.

**Funding:** This study was funded by the Department of Population Health of the medical college at Aga Khan University and International Organization of Migration -(UN Agency).

**Competing interests:** No authors have competing interests.

made available to poor refugee communities. Future research could explore which measures for disseminating factual information work best in refugee populations with different cultural norms and how best to target interventions to these groups.

## Background

Across the globe, concerns have developed that governments have failed to address the needs of refugees in dealing with COVID-19 since the virus was identified. The global health community has been particularly worried about refugees' heightened risk for COVID-19 infection due to their high mobility and living conditions that make it harder to adopt and adhere to preventive measures such as social distancing, wearing masks, self-isolation and regular hand washing [1–5]. In addition to endemic poverty, lack of migrant-inclusive health policies and dense living conditions, refugees also face language barriers, social stigma, lack of access to water and basic sanitation and minimal access to health care services due to the lack of medical insurance and refugee-tailored, inclusive health facilities [2–5]. Furthermore, the majority of refugees living in sub-Saharan Africa are hosted in countries with weak health and social protection systems [4, 5].

Even before the onset of COVID-19, studies reported disease outbreaks across many refugee and migrant centres due to the dire conditions. Modelling studies predicted increased infections in refugee communities [6, 7]. A modelling study in Bangladesh, for instance, indicated that steep investments in health care capacity and infrastructure would be required to combat the potentially disastrous impact of the COVID-19 epidemic in one refugee settlement, but given Bangladesh's fragile and overburdened economy, such an investment was not possible [7].

The COVID-19 pandemic has worsened the economic situation of urban refugees, many of whom were already living below the poverty line before the pandemic [4–7]. Furthermore, equitable health service delivery is hampered by refugees' tendency to disperse throughout urban areas [4, 5]. Refugee women, who frequently have limited access to maternal health services, are more likely to experience poor pregnancy outcomes. All these conditions persisted during the COVID-19 pandemic [1, 8, 9], which clearly exacerbated the existing health care gap for refugee women—specifically perinatal refugee women residing in urban Kenya.

Little empirical evidence is available on how COVID-19 affected refugee communities and particularly refugee women, what they know about COVID-19, what they do to prevent infection and the role of health care workers. In this paper, we report on refugee women's knowledge, their understanding of COVID-19 and key influences on compliance to public health recommendations. This is the second in a series of papers from our study that explored the impact of COVID-19 on women's access to antenatal and postnatal services and their knowledge and understanding of COVID-19 preventive measures. Our first paper from this study reported on the impact of COVID-19 in increasing women's preference for home deliveries [1]. We hope these papers will shed light on some of the issues faced by refugees living in low-resource settings, guide policy and exploration through future research.

## Methodology

### Social and cultural context

This study was conducted in Eastleigh, a semi-cosmopolitan centre in Kamukunji sub-county, Nairobi, in October 2020. Eastleigh is a vibrant business hub with a rich socio-cultural mix of people from the countries of the horn of Africa: Kenyans, Somalians, Eritreans, Ugandans,

Tanzanians, Ethiopians, Rwandese, Burundians and Sudanese. The majority of the population in this area are from Somalia, are of the Muslim faith, and have lived in the area for many years, some even decades.

Following the detection of the first COVID-19 cases in Kenya in March 2020, the Kenya Ministry of Health devised lockdown measures which deterred the movement of people to and from Nairobi. Due to dense population and high refugee mobility in the Eastleigh area, COVID-19 infection spread fast, and the government prohibited movement to and from Eastleigh for one month, from 6 May to 6 June 2020. While this lockdown was intended to stop rapid infection of COVID-19 to other parts of the city, by and large these measures infuriated the residents of Eastleigh as they considered them to be marginalising due to their vulnerable refugee status. The lockdown led to delayed care as well as an increase in women preferring home delivery of babies [1]. By the time this study was conducted in October 2020, Eastleigh had resumed business activities. The socio-cultural and economic context of Eastleigh is documented in more detail in our first paper from this study [1].

### Study design

This was a rapid qualitative study employing in-depth individual interviews [IDIs].

### Aim

This paper reports our findings on the knowledge, understanding and influences on compliance with COVID-19 preventive measures identified by antenatal and postnatal refugee women and their health care workers.

### Study population and sampling methods

To broaden and deepen our comprehension of how COVID-19 impacted access to and utilization of health care services, we conducted in-depth interviews [IDIs] with three facility-based health care workers [FHCWs], seven community health volunteers [CHVs] and 15 refugee women receiving antenatal care services [ANC] [n = 10] or postnatal care services [PNC] [n = 5]. Facility Health Care workers were recruited for this study because they play a central role in keeping records at the International Organization for Migration [IOM] clinic in Eastleigh that specifically serves the refugee community. Their inclusion was based on the premise that they could provide vital information on the refugees' attendance and uptake of services at the facility during the early periods of the pandemic as well as provide some information on what measures were undertaken at the facility to help staff and patients comply with COVID-19 preventive measures. Community health volunteers (CHVs) who served the refugee community were deemed an appropriate population for exploring their observed experience working with refugees in the communities and linking them to the facility services. Table 1 below illustrates the study sample characteristics.

### Data collection

We conducted data in October 2020 after ethical approvals were obtained from the Aga Khan University, Kenya Institutional Ethics Review Committee [Ref: 2020/IERC-84 (V2)] on 3 August 2020. Research was licensed by the National Commission for Science, Technology and Innovation [NACOSTI/P/20/6507] on 11 September 2020. Further approval was sought from Nairobi Metropolitan Services [Ref: EOP/NMS/HS/7/VOL.1/RS/30] on 5 October 2020.

Considering the diversity of the refugees, the study guide was developed in both English and Kiswahili, with separate components for the health care workers, antenatal women and

**Table 1. Summary of the study sample and population description.**

| Categories | Total number of participants interviewed | Description of participants |
|---|---|---|
| Facility health care workers | 3 | • More than one year of experience working with refugees |
| | | • Experience working with maternal child health services, such as antenatal, postnatal and HIV patient care |
| | | • Provided services to the communities including counselling |
| | | • Native Kenyans |
| Community health care workers | 7 | • More than one year of experience working with refugee community |
| | | • Facilitated antenatal and postnatal referrals from the refugee communities |
| | | • Six of Somali origin and Muslim, with the exception of one CHV who was a native Kenyan |
| | | • Roles were varied, including linking patients from the community to facility care, following up on contact tracing for tuberculosis patients, conducting household disease surveillance, promoting community hygiene and facilitating antenatal and postnatal uptake of services |
| Antenatal patients | 10 | • Participants by nationality: Somalis (n = 6), Tanzanians (n = 2), Ugandans (n = 1), Eritreans (n = 1) |
| | | • Had more than three children; the largest family had 10 children |
| | | • Aged 19 to 29, all unemployed, had lived in Eastleigh for at least one year and were married |
| Postnatal patients | 5 | • All were Somalis and had delivered during the pandemic |
| | | • Minimum number of children in the family was two and highest was nine |
| | | • All married, unemployed, of Muslim religion and lived in Eastleigh |

postnatal women. First, the research team from IOM and the Department of Population Health (DPH) at the Aga Khan University (AKU) met regularly for 4 months to develop the interview guides. Teams of qualified social scientists further reviewed the study tools to identify gaps. Secondly, following the review of the English version of the tools, the guides were given to IOM-qualified translators with many years of experience working with the refugee communities to translate the guide from English to Kiswahili. Upon completion of the translation, the questions were reviewed by the AKU principal investigator (AL), and an initial pilot of the guide was conducted with a health care worker and an antenatal clinic patient who were not included in the final study. Concerns raised included, for instance, the need to simplify the complex meaning of some words, such as *antenatal*, for the refugees. We therefore decided that, during the interview, a Somali translator would be available before the start of all interviews, and any of the refugee participants needing to converse in any of the Somali languages would be supported. Lastly, the principal investigator trained the research assistant and a part-time qualitative consultant who conducted some of the interviews. Issues around COVID-19 preventive measures and adherence during field work were discussed, and we impressed upon the research team the importance to adhere to COVID-19 preventive measures during data collection. For example, the research team had available face masks and hand washing gel which they provided to all participants before the start of the interviews.

Our participants were recruited by staff working at the Eastleigh clinic who had direct links with all the CHVs and postnatal [PNC] and antenatal [ANC]women in the communities. The study principal investigator provided information about the study to the recruiting team,

which they explained to potential participants before accepting them into the study. The recruitment of ANC and PNC women occurred at the IOM clinic during their weekly appointments, and they were given a time and day for the interview. Some participants were recruited through phone calls from FHCWs at the IOM clinic.

During the interview, the research team supplied study information to the participants, including the importance of adhering to COVID-19 preventive measures during the interviews, the objective of the study, the importance of confidentially, voluntary participation, benefits and possible risks involved and what they should do should they feel uncomfortable. Information sheets and informed consent, available in both English and Kiswahili, were provided in participants' preferred language. Interviews began after participants provided informed consent. Because some of the participants were illiterate, they signed consent forms using a thumb pad with the help of the study translator.

Five of the 15 refugee women interviewed did not speak Kiswahili or English, so we worked with an interpreter who was a trained community health volunteer directly linked to the Eastleigh Clinic. The IDIs for the FHCWs and CHVs were led by the study principal investigator, a Kenyan social scientist with more than 16 years' experience in research, while the IDIs with refugee women were conducted by a trained qualitative researcher and a research assistant. The IDIs were held in private clinical rooms within the clinic, which was considered a convenient and confidential environment for the interviewees.

Data collected included participants' socio-demographic information, knowledge and understanding of COVID-19, what training had been offered and by whom, compliance with COVID-19 preventive measures, challenges faced in complying with COVID-19 measures and possible enablers to compliance with public health recommendations, among other data.

## Data management and analysis

All audio-recorded interviews were transcribed verbatim from Kiswahili to English by a professional consultant and uploaded to a Department of Population Health password-protected laptop. The research team (AL and SO) listened to the audio-recorded transcripts and compared them with the transcribed data and reflective notes developed during the interviews. We then anonymised the transcripts by deleting references to names and roles and uploaded them in NVivo 12, a qualitative data analysis software program.

We first reviewed three transcripts, developed codes separately and then compared codes and developed the initial coding framework that was used to code the remaining transcripts. Initial codes that we developed were charted into categories and then thematic areas as shown below. A Consolidated Criteria for Reporting Qualitative Studies (COREQ) checklist, S1 Appendix, was used in reporting our study findings.

## Results

### 1. Refugee women's knowledge about COVID-19

Refugee women participants for this study were asked to share their experiences regarding their current knowledge, attitudes and perceptions towards COVID-19. In particular, they were asked what they knew about COVID-19 and associated health risks and what information they had received. Refugee women demonstrated an awareness, and understanding, of COVID-19 preventative measures.

*Washing of hands, wearing of masks. Yes, one must wash hands properly, regular sanitizing of hands and one has to wash their hands when they get back home.*

***IDI–refugee attending PNC 4***

*We were sensitized [by CHVs and IOM staff] that we should not walk along the roads with the child, we avoid being close to each other, we should not get out without a mask, we shouldn't even stay without washing hands, we avoid shaking of hands.*

**IDI–refugee attending ANC 5**

Some refugee women were aware of the symptoms and risks associated with COVID-19. Awareness of the severity of the disease was also very high, with some of the participants describing COVID-19 as 'a deadly disease' and 'kills directly'.

*It is a disease which comes and kills directly. Signs and symptoms are coughing, flu.*

**IDI–refugee attending ANC 6**

*A person with corona presents with body fever and cough mixed with fluid like mucus. That is what corona is.*

**IDI–refugee attending ANC 3**

*Corona is deadly disease, according to what I have heard, it can kill people.*

**IDI–refugee attending ANC 1**

Overall, we found a universal awareness of COVID-19 and preventive measures among the refugee women, indicating that their knowledge is appropriate. The findings indicate that wearing masks, hand washing, social distancing, avoiding handshakes and restricted movement were the most commonly identified prevention and control measures. The narratives also demonstrate that the government and CHVs had provided ongoing sensitization about COVID-19 among the urban refugee communities. However, only some of the women could identify a few of the symptoms of COVID-19.

## 2. Refugee women's sources of information about COVID-19

Sources of information about COVID-19 ranged from the local news (television and radio) to informal networks and the CHVs' outreach services.

**Local news as a source of COVID-19 information.** The local news media, particularly radio and television, and healthcare workers in the local hospitals in which they visited were identified as sources of information about COVID-19 and associated prevention and control measures.

*I heard it over the radio. When I listen to the news over radio, you are supposed to protect yourselves, you don't walk around or to sit in groups, you must follow instructions. When you go to the shops you must have your masks.*

**IDI–refugee attending ANC 4**

*I heard from hospital and even from news. From the television, also, I have heard it even from the government hospital when I visit there.*

**IDI–refugee attending ANC 5**

**Informal networks.** Informal social networks and word of mouth in residential areas were also primarily identified as authoritative sources of information on COVID-19, with newspaper and television cited as complementary sources.

*I heard from other people that there is a disease from China, then after while it arrived in Kenya but we took it lightly. . .. It is true; it is even common in our estate to find people seated together in groups as they chat and say that there is no corona.*

**IDI–refugee attending ANC 4**

**The Role of CHVs as a source of information for refugee communities.** Although the refugee women reported that their main sources of information were the media and their informal networks, narratives from the CHVs working with the refugee communities indicated that they also played a significant role in educating the community about COVID-19 preventive measures and symptoms. Detailed findings on how refugees delivered education about COVID-19 is reported in detail in our separate paper focusing exclusively on this communication. In the following quotes, some CHVs explain how they helped families reluctant to follow preventive measures.

*Moderator: Do you think the training you received on COVID-19 helped you?*

*Respondent: Yes, because this has also helped our community as you would find people with small children never bothered to wash their hands, and we encouraged them to wash their hands after every 10–20 minutes and ensure they take care of their children, they avoid crowded places and, if they must, they should wear a mask. We advised them that it was not safe for a pregnant woman to contract corona.*

**IDI–Community Health Volunteer 2**

CHVs applied various approaches, including the use of megaphones and designated vehicles, for wider coverage in the refugee community.

*After being trained we were to pass the information to the community. We even moved all over Kamukunji sensitizing people using a [megaphone] to make sure they get the information properly. We could use a vehicle rotating, we could stop to educate people on prevention and control measures. It was like we were TOTs [trainer of trainers].*

**IDI–Community Health Volunteer 3**

Further, CHVs provided telephone numbers for the refugee women and their families experiencing COVID-19 signs to call in case of emergency. Additional efforts were made to provide simple information using posters as well as translating the information into languages well understood by refugees.

*Respondent: The training I was given regarding COVID-19 is that I was to educate my community on what COVID-19 is, the symptoms and where they can get assistance and the number to call just in case there is a person in the household who has those symptoms.*

**IDI–Community Health Volunteer 5**

*After receiving the information, we dedicated time and explained to people, because there are those people, especially refugees, that don't want to know about these facts. They claimed it was not possible for them to observe the social distance so it was difficult for us at first, but when the numbers spiked up, they started getting convinced.*

**IDI–Community Health Volunteer 4**

These quotes illustrate how empowering CHVs with training on the early onset of epidemic can be an efficient in the dissemination of knowledge and information among marginalized people such as refugees.

## 3. Influences on Refugee Women's Compliance with COVID-19 preventive measures

Findings predominantly from CHVs revealed that in the 2 months following the first COVID-19 diagnosis in Kenya, most refugees did not believe that COVID-19 really existed, and by the time this study was conducted (October 2020), some community members still doubted that COVID-19 was a reality. Political and religious factors, fear of contracting COVID-19 and non-compliance with COVID-19 preventive messages and measures were mentioned as contributing to the spread of the disease.

**Political influences.** Findings largely from HCWs and CHVs showed that local and international political narratives concerning COVID-19 might have had a negative bearing on how refugees first reacted to and complied with COVID-19 measures. Refugee women did not trust government messages about COVID-19. They believed that information about COVID-19 was created for political gain and therefore saw no need to follow the guidance on using masks when attending the health facilities. Some asked for evidence of people dying from COVID-19.

*What they were telling us [refugees], if at all there is corona, let the government say the person or show the picture of that person who has died because of corona. That was [their] response.*

*IDI–Community Health Volunteer 5*

Because some believed that COVID-19 guidelines were politically motivated, they did not take mask wearing seriously and only carried a mask in case they encountered police.

*Some of them could come in [to the health clinic] even without a mask. When you ask them, they say we don't have corona, our politicians are using that excuse to get money. That is how they came in with information. Some of them could be having the masks, just hanging it but not using it effectively because they say if I don't have a mask the policeman will catch me outside there, so I will have to keep it here, if I see them, I just put it on.*

*IDI–Facility Health Community Worker 1*

Findings indicate that information given to refugees about preventive measures conflicted with international news suggesting that wearing of masks was equivalent to inhaling carbon monoxide. Conflicting messages may have made it difficult for refugees to change their behaviours considering, for instance, that some of these statements were coming from the siting United States president. Importantly, these findings may indicate how quickly marginalised communities can be influenced by politicians.

*Moderator: Do you think there is any information gaps that needs to be addressed or communicated to the refugees?*

*Respondent: There is big gap. It seems like they forgot everything they were taught. They need to be reminded every time. Sometimes they watch news, there was a debate in US [presidential debate] they heard [from the siting president] that if you are using a mask you are inhaling*

*carbon monoxide, it is affecting you [more] than helping yourself. So, some of them don't even use that mask because of that information that came out.*

***IDI–Facility Health Care Worker 2***

In general, and in contrast to the narratives from the refugee women, HCWs and CHVs reported that refugees did not follow guidance on wearing masks and held misconceptions about COVID-19, including, for instance, that it was a product of politicians' imaginations to gain political mileage. The US presidential debate also convinced some that masks would cause them to suffocate. The perspectives of FHCWs and CHVs indicated the level of misinformation about COVID-19 and a lack of compliance with COVID-19 guidelines among refugee women.

**Religious influences.** The study data suggest that, predominantly Muslim refugee communities initially believed that COVID-19 could affect only Christians as a punishment from God and that Muslims would be protected if they continued to pray five times a day. In the following quote, a CHV working with both the host and refugee communities explains how religious beliefs may have influenced compliance with COVID-19.

*Moderator: What I am wondering apart from the political influence you've talked about, are there other reasons that made the refugees not to believe COVID-19 existed?*

*Respondent: They always say that God produced that corona virus to punish the Christians, not the Muslims. . . . There are some rumours that went spreading that corona is not for Muslims and that it is for the Christians and that one should keep on praying five times a day and you won't have corona.*

***IDI–Community Health Volunteer 6***

Further, religious factors also shifted perceptions about COVID-19. For example, researchers found that the death of a well-known imam from COVID-19 might have caused Muslim refugees to shift their perceptions about 'COVID-19 not being real' and consequently to improve their compliance with preventive measures.

*At first, they [refugee community] were so negative but after and when we continuously explained to them and did awareness [about COVID-19 preventive measures] . . .there was one refugee, who is their imam, who died of COVID-19 in the mosque. . ., this made refugees realize that it was serious. How the mode of burial was conducted and the COVID-19 protocols made the people's mentality changed and they became positive [began to comply to preventive measures].*

***IDI–Community Health Volunteer 4***

*Respondent: Finally, they had to see the reality of it.*

*Moderator: What changed that?*

*Respondent: After hearing that there was imam who passed away, then they realized there's corona and they took a turn to take precautions. . ..*

***IDI–Community Health Volunteer 5***

Religious leaders across Africa in many denominations wield a lot of influence on their congregants. In many parts they are expected to provide leadership including provision of timely

information. As we have seen, the death of the imam played a huge role in shifting perceptions among the refugee population as they could then identify someone in power, they could relate to who was actually affected by COVID-19. This may show that the need for tangible evidence among communities is real, and documenting living stories, experiences and views of community influencers is an important aspect of risk communication. When no evidence is available of the disease following its course and resulting in severe death or illness, communities may find it difficult to accept the stated impacts.

**Family size and economic challenges.** The data suggest that refugee family size and the economic challenges they faced might have influenced whether they adhered to COVID-19 measures. The findings indicate that the refugees tend to have families of up to 15 members residing in a single household, which can impede social distancing and mask wearing. The inability to afford masks and soap for hand washing was also cited as a barrier to compliance across all categories of participants.

*In the house they are maybe more than ten to fifteen members. And I do not expect that if they are at home, they will be using the masks. So that is the problem because if it was social distance and they are so many on the house and they don't use the mask, I don't think they could come out of the house and practice that. So, they carried that kind of social life from their own families to the community.*

*IDI–Facility Health Care Worker 2*

*Some of them when they came to the facility and when asked at the entrance about a mask, the answer was, we don't have money to buy. That barred them from seeing a doctor, as they could not access a doctor without wearing a mask, and such refugees returned home without receiving services.*

*IDI–Community Health Volunteer 1*

The findings indicate the impracticality of control measures in households characterized by a large number of family members where social distancing may be difficult to practice. Further, many of the refugees could not afford masks and soap due to their low economic status.

## Discussion

Our findings show that refugee women are aware of COVID-19 symptoms and of globally accepted measures to prevent infection. The information has been widely disseminated through television, radio and CHVs' outreach services. The findings confirm that, in marginalized communities, CHVs play a key role in disseminating information. They also confirm the finding of earlier studies that, in low- and middle-income countries (LMIC), CHVs who reach out to marginalized communities can significantly increase their use of public health services [9].

Our study further revealed how misinformation spread by religious and political leaders has deeply affected refugees' beliefs, attitudes and behaviour with regard to COVID-19. Religious leaders propagated and perpetuated the misconception that Muslims are immune to COVID-19, and this idea was strongly endorsed by many followers. That the death of a well-known imam from COVID-19 helped shift notions about the virus among Muslim refugees underscores the crucial role religious leaders across the African continent could play in bringing about beneficial changes in behaviour. It demonstrates the need for public health practitioners and Ministry of Health staff to enlist the help of religious leaders in the fight against COVID-19. Studies have shown that in patriarchal communities, male engagement in

promoting behaviour change is key for the success of interventions [10]. We suggest that in marginalised communities', male religious leaders such as the imams, bishops and priests could be trained in preventive interventions and given simple messages to disseminate to their congregants.

Politicians have also greatly affected refugee communities' desire to comply with recommended COVID-19 prevention measures. The baseline position of most refugees—who have a strong ingrained mistrust of government that may be based on experience—was that COVID-19 was a ploy created by the government. Political misinformation also derailed compliance. 'Fake news' circulating locally and abroad played a significant role in influencing the refugee population. This and other forms of misinformation created confusion and left refugees sceptical about the efficacy of preventive measures and, therefore, the need to adopt them.

Our interviews show how such misinformation has undermined adherence to recommended COVID-19 prevention measures. Because of political officials' authority in the community, their posture on health issues exerts a clear effect on public behaviour. During a pandemic, marginalized and vulnerable communities, which may not have access to varied sources of information, can be highly influenced by political rhetoric.

This finding underscores the need to strengthen health care and prevention programs in marginalized communities and to empower them with factual information. In addition to engaging men and religious leaders to help get the message across, the programs need to work with CHVs to promote beneficial behaviours. In addition to the work predominantly conducted by female CHVs in this community, behaviour change can also be brought about by having other key influencers in Muslim refugee communities—such as mothers-in-law [11], grandmothers and male CHVs—disseminate messages about COVID-19.

Women refugees are greatly affected by poverty. To them, getting by on a daily basis required a steep effort even before COVID-19. Lack of jobs and other sources of livelihood made it very difficult to purchase masks, water, soap and sanitizers, which play an essential role in prevention at the community and family levels. Poverty was amplified by large family household arrangements characteristically exhibited by the refugee communities, making it difficult to maintain social distancing and purchase masks, water and sanitizers. Besides these barriers, however, the awareness and sensitization programs carried out by the CHVs and HCWs improved awareness and understanding of the disease.

## Recommendations

Although these findings are from a small group of refugees, primarily of Somali origin in Eastleigh, Nairobi, they provide some insights that may be applicable to refugee communities living in other resource-constrained settings. Local governments need to work closely with organizations serving refugees and migrants, as well as partner with the World Health Organization to initiate tailored trainings and capacity building among refugee communities and their health care workers.

We have seen the role religious beliefs and political rhetoric play in influencing both positive and negative compliance to COVID-19 preventive measures; therefore, religious and political leaders should be involved in mass training and public health messaging. Developing relevant behaviour change materials for refugees may require a focused effort due to their high mobility, mistrust and suspicion of the host countries and their reluctance to change. Local governments and the United Nations International Organization for Migration should work together and increase funding and the health care work force to work patiently with refugees in emergencies such as the COVID 19-pandemic to shift some of their behaviors.

Our findings suggest a need for more research to test refugees' knowledge and compliance with COVID-19 preventive measures, as well as the mode of messaging that works best for them, and to determine who can disseminate these messages most effectively. Given the confusion at the onset of COVID-19 about whether it was politically instigated, comparable studies should be conducted in other refugee settings and non-refugee settings to establish to what extent religious and political inclinations influenced compliance.

## Conclusions

This study suggested that awareness of COVID-19 is near universal among refugee women living in urban Kenya. Evidence gleaned from our research indicates an urgent need to develop and disseminate targeted guidelines and programs to help refugee women and their families in Kenyan cities prevent the spread of COVID-19. Future research could explore measures for disseminating factual information that work best in refugee populations with different cultural norms, help to hone targeted interventions and poll key stakeholders to find out what the community needs to protect itself from COVID-19 infection.

## Study limitations and strengths

This study's limitations include a small number of participants, the majority from a highly specific population (Muslim refugee women accessing ANC and PNC services) and language barriers that may have resulted in loss of vital information through translation. Despite these limitations, this study improved our knowledge of the experiences of refugee women and their families in the early periods of the pandemic and deepened our understanding of key influences on compliance with preventive measures.

## Supporting information

**S1 Appendix. Consolidated criteria for reporting qualitative studies (COREQ): 32-item checklist.**
(DOCX)

## Acknowledgments

We are grateful to the Department of Population Health of the Medical College at the Aga Khan University, Kenya, and the United Nations International Organization for Migration staff who volunteered their time to conduct this study. We are also grateful for the participation of the Eastleigh Wellness Clinic team, research participants and field researchers. Lastly, we would like to thank Carol Jaka for initial review of the tools and manuscripts, for the team from Institute of Human Development (IHD) Aga Khan University and Sheffield University for reviewing and providing comments on the study tools.

## Author Contributions

**Conceptualization:** Adelaide M. Lusambili, Michela Martini, Asante Abena, Stanley Luchters.

**Data curation:** Faiza Abdirahaman, Asante Abena, Joseph N. Guni.

**Formal analysis:** Faiza Abdirahaman, Joseph N. Guni, Sharon Ochieng.

**Funding acquisition:** Adelaide M. Lusambili, Michela Martini, Stanley Luchters.

**Investigation:** Adelaide M. Lusambili, Asante Abena, Sharon Ochieng.

**Methodology:** Adelaide M. Lusambili, Michela Martini, Faiza Abdirahaman, Asante Abena.

**Project administration:** Michela Martini, Faiza Abdirahaman.

**Resources:** Michela Martini, Stanley Luchters.

**Supervision:** Stanley Luchters.

**Validation:** Adelaide M. Lusambili, Michela Martini, Faiza Abdirahaman, Stanley Luchters.

**Visualization:** Asante Abena, Stanley Luchters.

**Writing – original draft:** Joseph N. Guni, Sharon Ochieng.

**Writing – review & editing:** Asante Abena, Stanley Luchters.

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
