## [Decision Letter · Decision Letter 0]

23 Jul 2021

PONE-D-21-14476

Knowledge, Understanding and Socio-Cultural Influences to Refugees Compliance to COVID-19 Preventive Measures: A Qualitative Inquiry of Health Care Workers and Refugee Women attending antenatal and postnatal health services in Urban Kenya.

PLOS ONE

Dear Dr. Lusambili,

Thank you for submitting your manuscript to PLOS ONE. After careful consideration, we feel that it has merit but does not fully meet PLOS ONE’s publication criteria as it currently stands. Therefore, we invite you to submit a revised version of the manuscript that addresses the points raised during the review process.

We look forward to receiving your revised manuscript.

Kind regards,

Susan A. Bartels, MD, MPH, FRCPC

Academic Editor

PLOS ONE

Journal Requirements:

2. When reporting the results of qualitative research, we suggest consulting the COREQ guidelines or other relevant checklists listed by the Equator Network, such as the SRQR, to ensure complete reporting (http://journals.plos.org/plosone/s/submission-guidelines#loc-qualitative-research). In this case, please consider including more information on the recruitment of participants. Moreover, please provide the interview guide used as a Supplementary file.

4. Please amend the manuscript submission data (via Edit Submission) to include author Sharon Ochieng,and Stanley Luchters.

5. Please amend your authorship list in your manuscript file to include author Constance Shumba.

Reviewers' comments:

Reviewer's Responses to Questions

**Comments to the Author**

1. Is the manuscript technically sound, and do the data support the conclusions?

Reviewer #1: Yes

Reviewer #2: Yes

2. Has the statistical analysis been performed appropriately and rigorously? 

Reviewer #1: N/A

Reviewer #2: I Don't Know

3. Have the authors made all data underlying the findings in their manuscript fully available?

Reviewer #1: Yes

Reviewer #2: Yes

4. Is the manuscript presented in an intelligible fashion and written in standard English?

Reviewer #1: Yes

Reviewer #2: Yes

5. Review Comments to the Author

Reviewer #1: Thank you for the opportunity to peer review this important piece of work. This qualitative inquiry of health care workers and refugee women in urban Kenya aimed to report findings on knowledge, understand and influences to refugees compliance to COVID-19 preventive measures. The authors used in-depth individual interviews with 10 health care workers/volunteers and 15 refugee women to explore their knowledge, understanding and influences to compliance with COVID-19 IPC. This work highlights the high level of awareness and prevention control measures among refugee women, together with poor level of compliance and goes on to explore what may be underpinning this. This is an important piece of original work, with an appropriate study design and ethics approval, carried out during a pandemic with a vulnerable group in urban Kenya and the findings could inform future pandemic preparedness plans.

I believe the manuscript requires minor editing, and I note some suggestions below.

I would welcome further details on the methodology, particularly on the social cultural context as this is key to explain what may underpin refugee women’s knowledge, understanding and influences. If possible, it would be important to know about the socio-demographics profile of the participants in the study e.g., family size, socio-economic status etc and this should be presented with the results.

Regarding the aim (line 77), it is important to justify what the added value is of also interviewing FHCWs and CHVs as well refugee women – this could be detailed in the introduction or the methods section. Finally in lines 81-86 please ensure you are consistent with your writing style and use either numbers or words to describe the number of participants sampled. Building on that, I would welcome further details around the context of COVID-19 in October 2020 in this area of Kenya. This will help to make a judgement of what the knowledge of the study participants should be, it will also be useful to know about any public health communications campaigns that may have been carried out on this topic.

Overall, the conclusions made are supported by the results presented. However, it would be valuable to discuss the results further considering previous evidence around compliance of IPC measures and refugee groups for example.

Finally, I think it would be important to have a paragraph discussing future policy implications of your findings and recommendations to policymakers, especially in view of your findings that highlight the impact of pollical factor on refugee’s compliance to COVID-19 preventive measures.

This manuscript is important for the field of migrant health and pandemic preparedness and the qualitative findings can inform future health policy and planning.

Reviewer #2: PLOS Peer Review Jul 2021 (DRSMS)

Knowledge, Understanding and Socio-Cultural Influences to Refugees Compliance to COVID-19 PReventative Measures: A Qualitative Inquiry of Health Care Workers and Refugee Women attending antenatal and postnatal health services in Urban Kenya. Lusambili et al

Line159 - 166

This is comparable to our experiences in Camp Moria in Lesvos where 18000 people were residing in extremely unhygienic and overcrowded. As the main primary medical provider inside the camp during the early months and throughout the pandemic, we observed much-reduced numbers in the clinic as people lived fearfully of contracting this fatal illness. Conversely, the authorities and the local Greeks assumed that the refugees were unaware and chose to live complacently. The truth is - they had very little choice- often not even running water. This great sense of fear led the refugees to believe that their lives did not matter and that they would be left to die.

Line287

If this is indeed true...that Trump could even influence even these poor people- it’s a good thing that he is gone. However, I wonder if this sample size is too small to conclude anything like this.

Line325

Because of the small sample size again, it would be helpful to understand what religious faith group the refugees interviewed belonged to. Were they all Muslims? If not - what was the breakdown. It can come across as slightly biased. It may also be helpful to know the faith background (if any) of the HCW. Additionally, was it not Trump again who may have started the bigotted idea that ‘China was being punished’ at the onset of the pandemic- due to unconventional choices of food. The world was a confused place as the pandemic developed...until finally we all knew- this virus affected all equally. Something about this needs to be acknowledged in the discussion.

Line 383-391

Surely it was not just the refugees’ community who were confused by ‘fake news’ and misinformation and mistrusting political agendas. Many groups of people from across the globe suffered similar sceptiscm.

The role of healthcare worker advocacy and education in this regard should be stressed on more-> as HCW are in the best position to influence those they care for.

Line 393-397

Does this section seem to indicate that the local HCW were relatively powerless to overcome the political and religious rhetoric. Is there enough evidence for this conclusion based on the interviews in this study? Was there any indication that if health promotion activities were undertaken- which type did the women find most helpful eg leaflets/ one-to-one or group teaching sessions?

Line403-404

Again I am made to assume that the majority/ all of the refugees women interviewed were from the Muslim faith. Is this a fact? I think this is answered in 418-419- but it really would be helpful to have this mentioned earlier in the paper.

Line 412-413

It would be good to see these bits elaborated. How did HCW/ CHV’s improve awareness? What did they do exactly?

Line 447

This point needs to be revised. Is there enough evidence from these small numbers to indicate that the refugee population alone was influenced by this? Surely the population at large (including the host country) would be affected by political and religious influences. Without comparative data on the host population- I would not agree that this research paper can conclude this.

The other two points are fine.

6. PLOS authors have the option to publish the peer review history of their article (what does this mean?). If published, this will include your full peer review and any attached files.

Reviewer #1: No

Reviewer #2: **Yes: **Dr Siyana Mahroof-Shaffi

---

## [Decision Letter · Decision Letter 1]

18 Oct 2021

PONE-D-21-14476R1‘ It is a disease which comes and kills directly’: What refugees know about COVID-19 and key influences of compliance with preventive measures.PLOS ONE

Dear Dr. Lusambili,

Thank you for submitting your manuscript to PLOS ONE. After careful consideration, we feel that it has merit but does not fully meet PLOS ONE’s publication criteria as it currently stands. Therefore, we invite you to submit a revised version of the manuscript that addresses the points raised during the review process.

We look forward to receiving your revised manuscript.

Kind regards,

Susan A. Bartels, MD, MPH, FRCPC

Academic Editor

PLOS ONE

Journal Requirements:

Additional Editor Comments (if provided):

Reviewers' comments:

Reviewer's Responses to Questions

**Comments to the Author**

1. If the authors have adequately addressed your comments raised in a previous round of review and you feel that this manuscript is now acceptable for publication, you may indicate that here to bypass the “Comments to the Author” section, enter your conflict of interest statement in the “Confidential to Editor” section, and submit your "Accept" recommendation.

Reviewer #1: All comments have been addressed

Reviewer #2: (No Response)

2. Is the manuscript technically sound, and do the data support the conclusions?

Reviewer #1: Yes

Reviewer #2: Partly

3. Has the statistical analysis been performed appropriately and rigorously? 

Reviewer #1: I Don't Know

Reviewer #2: I Don't Know

4. Have the authors made all data underlying the findings in their manuscript fully available?

Reviewer #1: Yes

Reviewer #2: Yes

5. Is the manuscript presented in an intelligible fashion and written in standard English?

Reviewer #1: Yes

Reviewer #2: Yes

6. Review Comments to the Author

Reviewer #1: Thank you for addressing the peer review comments. I believe the revised manuscript reads better than previous drafts. The aims and objectives are clearer and the changes to the title and certain sections are welcomed. No further comments.

Reviewer #2: see atatched. I found the tracked section of the new docuement difficult to follow...hoep my commnets make sense.

7. PLOS authors have the option to publish the peer review history of their article (what does this mean?). If published, this will include your full peer review and any attached files.

Reviewer #1: **Yes: **Dr Behrouz Nezafat Maldonado

Reviewer #2: **Yes: **Dr S. Mahroof-Shaffi

---

## [Author Response · Author response to Decision Letter 1]

30 Oct 2021

Dr Adelaide Lusambili

Department of Population Health

Aga Khan University Medical University

P.O. Box 30270-00100. Nairobi, Kenya

Email: Adelaide.lusambili@aku.edu

28th/Oct/2021

PLOS ONE Editorial Office

Re: Revision and resubmission of manuscript tiltled: ‘It is a disease which comes and kills directly’: What refugees know about COVID-19 and key influences f compliance with preventive measures. 

Thank you for your letter and the opportunity to revise our paper 

The suggestions offered by the reviewers have been immeasurably helpful in improving this paper. Reviewers comments have been individually addressed – all marked red. 

All the co-authors have approved the revisions, and I am still the corresponding author. We have also uploaded both tracked and a clean revised manuscript to your online system. 

We hope the revised manuscript will better suit PLOS ONE and we are happy to consider further revisions. 

Sincerely

 Dr Adelaide Lusambili

A REBUTTAL LETTER 

PONE-D-21-14476R1

‘It is a disease which comes and kills directly’: What refugees know about COVID-19 and key influences of compliance with preventive measures.

Journal Requirements:

Thanks. I have checked the references and amended. I have removed 

1. Reference 10 because it is unpublished paper. It is indicated in the tracked copy

Additional Editor Comments (if provided):

Reviewers' comments:

Reviewer's Responses to Questions

Comments to the Author

1. If the authors have adequately addressed your comments raised in a previous round of review and you feel that this manuscript is now acceptable for publication, you may indicate that here to bypass the “Comments to the Author” section, enter your conflict of interest statement in the “Confidential to Editor” section, and submit your "Accept" recommendation.

Reviewer #1: All comments have been addressed

Reviewer #2: (No Response) 

Thank you.

2. Is the manuscript technically sound, and do the data support the conclusions?

Reviewer #1: Yes

Reviewer #2: Partly

Noted. We recognize reviewer 2s concerns. This is due to the small sample size, which we have acknowledged in our conclusions.

3. Has the statistical analysis been performed appropriately and rigorously? 

Reviewer #1: I Don't Know

Reviewer #2: I Don't Know. Thank you. This was a qualitative study.

4. Have the authors made all data underlying the findings in their manuscript fully available?

Reviewer #1: Yes

Reviewer #2: Yes

Thank you noted. ________________________________________

5. Is the manuscript presented in an intelligible fashion and written in standard English?

Reviewer #1: Yes

Reviewer #2: Yes

6. Review Comments to the Author

Reviewer #1: Thank you for addressing the peer review comments. I believe the revised manuscript reads better than previous drafts. The aims and objectives are clearer and the changes to the title and certain sections are welcomed. No further comments. 

Thank you for reviewing this manuscript.

Reviewer #2: see attached. I found the tracked section of the new document difficult to follow...hope my comments make sense. 

Thanks. I have addressed your questions in the attached PDF form as follows

UPLOADED REVIEWERS COMMENTS [Comments by Dr.Siyana Mahroof-Shaffi]

47

It is questionable whether high mobility is an issue during various lockdowns. Our experience was that the refugee camps were the first to become ‘locked’ with limited movement in and out of the camps. Elaborate on issues such as poor hygiene in facilities/ access to WASH facilities and probable overcrowding.

Ref: (https://via.hypothes.is/https://www.medrxiv.org/content/10.1101/2020.07.07.20140996v2#annot

 ations:Q3NC_sCNEeqwwYO0MegHbw).ef Tucker et al:

The refugees we interviewed were not living in facilities but rather in a semi-urban [EASTLEIGH} area that hosts refugees from the horn of Africa. While poor hygiene in facilities and access to WASH is a recognized issue in refugee populations, however, the purpose of this study was to understand their knowledge, information and sources of these informations and the effect of COVID 19 on the utilization of maternal and neonatal services.

111 (Study population)

Very Low numbers already acknowledged. It would be good to understand how this compares to the size of the population being studied. How many in total are in the area? Also men vs women?

The purpose of the study was to do a rapid qualitative study to provide to promote our understanding on what was happening among this vulnerable group. We did not do a statistical comparative study which would have required as to compare data across age, gender etc as you have requested. We did not interview men, our focus was PNC and ANC women and the healthcare providers.

In our paper, cited reference 1, we have noted under social cultural context that …” Most of the refugees are of Somali origin. The 2019 national census put refugees at 55% of the overall Kamukunji sub -county population (147,551 out of 268,276), of which 51% are men and 49% are women”. 

244

Typo= ‘visited” DONE

370-401

This section could be perceived as being quite biased. It is based on the commentary of three community workers. It would be interesting to know which faith background the workers come from? 

I agree that it contrasts with the interviews with the patients themselves were far less loaded. I would suggest taking this section out as it seems unnecessarily inflammatory. 

 In general, there has been a lot of blame COVID (eg US blaming China...then China laughing as US cases soared). The take home message instead could be that despite all our technological and scientific success- there has been no rational explanation for the onset of this pandemic. More than one faith followers have expressed the phenomenon of this being a sign of the ‘end of the world’ and so it seems a bit harsh to penalise just the Muslims. In UK for example, various communities had continued to gather (eg Jewish wedding/ secret parties amongst youngsters)...ultimately the focus has to be on ignorance or defiance which can affect any group of people and not limited to certain faiths only. 

Thank you.

• In the Table 1 in the manuscript I have indicated that all the CHVs were of Somali origin and Muslim by religion with an exception of one who was a native. Now, in the quotes, we cannot identify their religion because it is easy for the public to link these quotes to them. We have the responsibility to anonymize all the data and maintain confidentiality of our interviewees.

• I have reworded the sentences to be inclusive – that these biasness happens across all religions and that religious leaders have influence over their congregants. These are views that were reported in the interviews – and deleting firsthand information from CHVs considered insiders in their own communities invalidates the purpose of this research. Religious influences in Africa where 90% of the population is affiliated to a religion is real and cannot be ignored. All CHVs interviewed revealed how difficult it was to convince refugees in the early onset of COVID19 due to religious beliefs. These are important insights that should prompt further research that can provide future pandemic preparedness for different groups. 

517-518

I rest my case…! This study cannot conclude anything based on the small and select group of interviewees.

We have acknowledged the study limitation –however –important to note is that –qualitative studies usually have a small sample.

Thank you. I agree –however-as we have stated, being a rapid study in a population that is poorly studied and underrepresented in research, this study provides first hand insights that could be tested further in large studies in refugee context as well as non-refugee context of different religious affiliation to determine communities early experience during COVID 19.

7. PLOS authors have the option to publish the peer review history of their article (what does this mean?). If published, this will include your full peer review and any attached files.

Do you want your identity to be public for this peer review? For information about this choice, including consent withdrawal, please see our Privacy Policy.

Reviewer #1: Yes: Dr Behrouz Nezafat Maldonado

Reviewer #2: Yes: Dr S. Mahroof-Shaffi

---

## [Editor Report · Decision Letter 2]

1 Dec 2021

‘ It is a disease which comes and kills directly’: What refugees know about COVID-19 and key influences of compliance with preventive measures.

PONE-D-21-14476R2

Dear Dr. Lusambili,

We’re pleased to inform you that your manuscript has been judged scientifically suitable for publication and will be formally accepted for publication once it meets all outstanding technical requirements.

Kind regards,

Susan A. Bartels, MD, MPH, FRCPC

Academic Editor

PLOS ONE
---

## [Editor Report · Acceptance letter]

10 Dec 2021

PONE-D-21-14476R2 

‘It is a disease which comes and kills directly’: What refugees know about COVID-19 and key influences of compliance with preventive measures. 

Dear Dr. Lusambili:

I'm pleased to inform you that your manuscript has been deemed suitable for publication in PLOS ONE. Congratulations! Your manuscript is now with our production department. 

Kind regards, 

on behalf of

Dr. Susan A. Bartels 

Academic Editor

PLOS ONE